# Decline of German and rise of North-American hegemony in science: Insights from Nobel Prize nominations (Physics/Chemistry, 1901–1969)

Marie von der Heyden *, Thomas Heinze

Institute of Sociology, University of Wuppertal, Wuppertal, North Rhine-Westphalia, Germany

* mvonderheyden@uni-wuppertal.de

## Abstract

This paper examines the shift in global scientific leadership from Germany to the United States using data on Nobel Prize nominations in Physics and Chemistry from 1901 to 1969. Building on the theoretical frameworks of Ben-David and Hollingsworth, we explore how nomination patterns reflect global shifts in scientific hegemony. In the early 20th century, Germany dominated the Nobel nomination process, with its scientists frequently acting both as nominators and nominees. During the 1930s, however, the United States rose to prominence, becoming the leading force in global science. By the mid-20th century, American scientists constituted a substantial share of both nominees and nominators, reflecting the nation's emergence as a global research leader. Self-nomination trends align with this hegemonic transition. Despite their dominance in the nomination process during their respective periods of global leadership, neither country demonstrated a particular capacity to influence selection outcomes. Regression analysis reveals limited advantages for nominees from hegemonic nations and no consistent effects for nominators, illustrating the distinction between controlling nominations and shaping laureate selections. This study offers insights into the dynamics of scientific prestige and the relationship between national hegemony and institutional frameworks.

## 1. Introduction

The global distribution of scientific leadership is closely intertwined with worldwide shifts in political and economic power, reflecting societal changes more broadly. Historically, nation states and their academic institutions emerged as dominant producers of scientific and technological knowledge, securing their place at the forefront of scientific discovery and technical innovation [1]. However, national leadership in science, technology, politics or economic matters is contested. Over time, global power and hegemony may change, with scientific leadership migrating from one country to another in response to shifts in both national priorities and international influence.

**Data availability statement:** Data and replication files are available at the OSF repository: https://osf.io/dgau3/

**Funding:** The author(s) received no specific funding for this work.

**Competing interests:** The authors have declared that no competing interests exist.

Such transitions are not simply the consequence of internal scientific advancement; they are also shaped by broader social, economic, and institutional forces.

The Nobel Prize has long been recognized as the pinnacle of scientific achievement. Reliance on Nobel Prize data as a metric for assessing global scientific leadership is based upon a substantial body of precedent. Laureates are employed as a metric for evaluating the efficacy of national science systems [2], scientific capabilities of research organizations [3], or the appeal of sub-disciplines [4]. The Nobel Prize also serves as a benchmark for measuring the quality of universities [5].

Previous studies have utilized laureate counts as indicators of national scientific prestige. Of particular interest concerning the compilation of laureates are concentration processes with regard to national or institutional hegemonic positions. Since the majority of laureates comes either from Europe or North America (especially from Germany and the United States), studies often deal with a concentration at the level of nationality [6–8]. In addition, other studies have analysed the organizational composition of the Nobel population, focusing on different stages of laureates' careers, such as the organizations where they received their academic degrees, conducted their award-winning research or received their prizes [3,9–11].

The question of how closely the measurement of Nobel laureates aligns with scientific quality is a topic of considerable debate. In her seminal work, Zuckerman [12] argued that the Nobel Prize is awarded by a meritocratic process, designed to recognize the best candidates. However, she was careful to acknowledge limitations of the prize system. One such limitation is the restriction on the number of laureates, which excludes many researchers who may also merit recognition. Furthermore, Zuckerman observed that when multiple candidates of equal qualifications are available, specific contextual factors, such as nationality, personal relationships, or institutional affiliations, may exert influence on the decision-making process. Nevertheless, Zuckerman argued that the awarding of the Nobel Prize operates within a universalistic framework.

In contrast to Zuckerman [12], historians such as Crawford [13] or Friedman [14] emphasized anecdotal evidence to the contrary, suggesting the influence of personal networks and relationships. Crawford for example, interprets the close ties between directors of Kaiser Wilhelm Institutes (later Max Planck Institutes) and the Nobel committees as an indication of potential bias [13, pp.106–124]. In particular, Friedman emphasizes the lobbying efforts of key actors such as Svante Arrhenius, an influential member of the Physics Committee and an informal but powerful figure within the Chemistry Committee. Arrhenius allegedly used his influence to support people he trusted, such as Fritz Haber, who received the Nobel Prize a few years after becoming director of the Max Planck Society, while hindering his adversaries: the delayed recognition of Walther Nernst, Friedman suggests, was a result of Arrhenius's strained relationship with him [15, pp- 87–109; 180–184].

Although sociologists, such as Zuckerman and Merton [16], were aware of the possibility of violations in an otherwise universalistic system, they considered the occurrence and explanatory power of such violations to be limited. They acknowledged that norms are not always adhered to in practice. Yet, they viewed such

violations as instances with the potential to reinforce the violated norm through subsequent processes of self-affirmation. Consequently, the occurrence of deviant behaviour would not undermine the underlying norm; rather, it may inadvertently reinforce it.

This paper presents a novel approach by mapping global shifts in scientific leadership through the examination of Nobel Prize nominations. The nomination network, comprising nominators and nominees, may also reflect the hierarchies of authority and influence that define international scientific hegemony. As global scientific leadership shifts between nations, so does the distribution of Nobel nominations, indicating where the most influential scientific work is being conducted and which countries and research organizations are positioned as global leaders.

Our study takes a quantitative approach, but it is important to recognise that many valuable qualitative studies have already contributed significantly to our understanding of the nomination process in specific fields [4,17,18]. For instance, Seeman & Restrepo's research [19,20] provides valuable qualitative insights into the Nobel nomination process in chemistry, offering detailed case studies of both successful and frequently overlooked nominees, such as Christopher Ingold. Their work emphasizes individual nomination trajectories and highlights the role of professional networks and biases in shaping the selection process.

In contrast, our study utilizes a quantitative approach that analyses the entire set of available nomination data, allowing us to draw broader conclusions about geopolitical shifts and structural patterns in scientific recognition over time. While qualitative case studies offer rich context, they often cannot be generalized and may be influenced by other biases. By systematically analysing the complete dataset, we aim to provide a comprehensive, data-driven perspective on how nominations reflect and shape the evolving landscape of scientific excellence.

We advocate for a methodological approach that combines large-scale quantitative analysis with historical context to capture the underlying dynamics of Nobel Prize nominations more effectively. This approach aligns with recent research on social and cultural closure in Physics Nobel Prize decisions [21].

The nomination data of this paper provides a detailed and nuanced view of the broader network of individuals and research organizations involved in the recognition of scientific achievement, rather than merely focusing on the winners. In comparison to laureate recognition, the nomination process serves as a more immediate reflection of contemporary scientific prestige, as nominations occur as soon as contributions have already diffused into the scientific community and are actively recognized as prizeworthy. While laureates are often honoured for work conducted decades earlier [22], nominations capture active debates and include more recent advancements. Moreover, nominations encompass a broader pool of prizeworthy discoveries, including many outstanding contributions that ultimately do not receive the award. As Zuckerman [12] notes, the scientific elite is not limited to Nobel laureates but extends to the "41st chair" – the many researchers who meet the highest standards of excellence but are not among the select few awarded the prize.

By examining the networks of nomination and the research organizations and countries from which the actors involved originate, one can gain a more nuanced understanding of the global distribution of scientific influence. This approach is consistent with the view that prestige and networks are key factors in science [23].

Furthermore, an analysis of Nobel Prize nominations provides insights into the ways in which national and institutional contexts influence the nature of scientific achievements. The operation of science within a system of cumulative advantage [16] results in the concentration of prestige and resources in a limited number of elite research organizations. The nomination process serves to reinforce these dynamics, thereby perpetuating the dominance of established centres of research and the existing hierarchies. By examining the manner in which these patterns undergo change, it is possible to trace the broader evolution of scientific hegemony and the factors that drive changes in leadership within the global scientific community.

The main findings of our study confirm the striking shift in global scientific leadership from Germany to the United States, as reflected in the patterns of Nobel Prize nominations. This shift is evident in the distribution of nominations: from 1901 to 1933, Germany held a dominant position, while from 1934 to 1969, the United States took the lead. Remarkably,

more than 40% of all nominees during this period came from the United States, underscoring its emergence as a global scientific leader.

Moreover, self-nomination rates are closely aligned with broader hegemonic shifts. German scientists were more prominent in self-nominations before World War II (WWII), but their presence declined after the war, coinciding with the rise of American scientists. Interestingly, while certain countries dominated the nomination process, their ability to influence the final selection of laureates was less pronounced. Statistical analysis shows only marginal advantages for nominees and no clear effects for nominators, suggesting that securing nomination rights as well as nominations is not the same as controlling the outcomes.

The paper is organized as follows: First, we outline the theoretical framework and derive main hypotheses that guide our analysis. Next, we present data and methodology, detailing the dataset of Nobel Prize nominations and the statistical methods employed. The empirical results are then discussed, highlighting key trends and patterns that support or challenge our hypotheses. Finally, we conclude with a summary of the findings and a broader discussion of their implications for understanding scientific hegemony and the dynamics of academic recognition.

## 2. Theoretical Framework and Hypotheses

Early in the 20th century, a substantial shift in scientific leadership occurred, with the global centre of science migrating from Germany to the United States. This shift in hegemony, which was precipitated by political and institutional changes, resulted in a significant alteration to the landscape of scientific research. Germany, which had been the epicentre of scientific excellence in the 19th century, began to lose its leading role, while the United States, with its developing academic infrastructure and increasing investment in scientific research, rose to prominence. As Ben-David [24] observed, the capability of nations to achieve scientific eminence has been linked to other structural characteristics of their respective education and political systems. To illustrate, Germany's ascension to global scientific leadership in the 19th century was enabled by a decentralized and highly competitive public university system that permitted mobility and fostered innovation. This period was distinguished by German research organizations providing cutting-edge facilities, academic autonomy, and career opportunities, which contributed to the professionalization of science. Previously conducted in private laboratories, science started to be incorporated into well-equipped research organizations.

Ben-David's [24,25] findings resonate well with more recent historical evidence on particular technologies and economic sectors. For example, Murmann [1] shows that global competitiveness and industrial leadership of German chemical industry (in particular: dye production) in the late 19th and early 20th century was in large part due to the abovementioned university system with which companies, such as BASF, Bayer or Hoechst, cultivated long-term and productive relationships [1, Chp. 4].

The historical trajectory of German universities also highlights their crucial role in fostering scientific and technological progress over the long term: German universities had already established themselves as key drivers of knowledge production and industrial innovation by the mid-1800s, with measurable effects on economic growth. This deep-rooted connection between academia and industry not only contributed to Germany's leadership in sectors such as chemistry but also shaped the international perception of its scientific institutions [26].

Notwithstanding these advantages, internal structural limitations constrained Germany's capacity to adapt to emerging research domains. The hierarchical system of German universities, in which professors held significant control over their subordinates, proved resistant to reform. This conservatism, coupled with an aversion to embracing novel fields of study, prompted innovative research to seek support beyond the university system, specifically in research institutes of the Kaiser Wilhelm Society, founded in 1911 [25].

Despite these challenges, Germany maintained its scientific dominance into the early 20th century. The historical embeddedness of German universities in knowledge-driven economic expansion [26] further explains why their influence persisted in global scientific networks, even amidst structural rigidities and political disruptions. This was partly due to the

continued recognition of German universities and the continued use of German as a scientific language. It was also due to the considerable number of scientific leaders before WWI and the inertia of the international scientific community, which continued to favour German universities. The favourable view of German universities by visiting academics remained undisturbed by structural tensions or occupational uncertainties. However, the inflexible structure of the university system hindered the formation of dynamic scientific communities and restricted innovation [25].

By the early 20th century, the United States had begun to surpass Germany in terms of scientific leadership. This shift was facilitated by the flexibility of the institutional framework, which enabled US universities to enjoy greater autonomy and foster dynamic, competitive environments conducive to the development of new research fields. The North-American system, distinguished by effective leadership, faculty collegiality, research-based graduate education, and structured career paths, fostered a conducive environment for scientific advancement [2,25].

Hollingsworth [27] builds on Ben-David's work by emphasizing the role of institutional environments in shaping innovation. He argues that research organizations that operate within a national institutional environment that exerts little control, such as Rockefeller Institute (and later: University) in the United States, enjoy greater organizational flexibility and thus are more successful in generating scientific innovation. This flexibility enabled scientists at Rockefeller to conduct pathbreaking research, in addition to facilitative leadership that was willing to take risks and to secure funding for unconventional research projects.

The results of quantitative analyses corroborate the qualitative assessments put forth by Ben-David and Hollingsworth, providing evidence for the historical shift in global scientific leadership [28]. Specifically, they highlight the long-term accumulation of scientific prestige that enabled the United States to surpass Germany as the dominant scientific power [29,30], as well as the comparative advantage of nations whose institutional set-up fostered scientific innovation [31].

One factor that has contributed to the ascendance of the United States (and the protracted decline of Germany) in global scientific leadership is the migration of scientists. The available literature finds that scientists tend to migrate from countries with stricter bureaucratic control to those with lesser control levels [31]. This mobility is driven by researchers that seek more conducive research environments, as well as by forced migration due to factors such as political persecution, as during Nazi Rulership in Germany between 1933 and 1945.

It is important to acknowledge the fate of the numerous individuals who were compelled to flee or who were killed by the Nazi regime. Between 1933 and 1945, a considerable number of scientists and scholars departed from Germany, motivated by discriminatory legislation such as the Law for the Restoration of the Professional Civil Service, which targeted Jewish and other "undesired" groups within the civil service, including university personnel. In fields like chemistry, physics, and mathematics, up to 18 percent of professors were dismissed between 1933 and 1940. Notable universities, such as those in Göttingen and Berlin, were particularly affected, with losses of personnel amounting to 60 percent [32,33].

Thus, the migration of researchers resulted in a substantial alteration of the student-to-faculty ratio at leading universities, which had a negative impact on the quality of teaching and the calibre of faculty. Furthermore, the loss of highly productive and prestigious scientists, in particular, led to a long-term decline in German publication output [34]. Conversely, the influx of German emigrants to the United States contributed to the advancement of US American science, as evidenced by an increase in patenting rates within fields where scientists working in Germany had previously dominated [35]. In light of the existing literature on the historical shift, we put forward the following hypothesis:

H1: Nominations for the Nobel Prize come disproportionately from scientists whose research organizations are located in nations that wield global scientific hegemony.

The intersection of nationalism and internationalism in the scientific system is a central theme in Crawford's [13] analysis of nomination clusters during and after World War I (WWI). She posits that these clusters provide evidence that nominations are influenced by particularistic factors, such as nationality. Crawford identifies the phenomenon of "own-country" nominations, whereby approximately 50% of nominations in major countries (France, Germany, the UK, and the US) are for scientists from their own country. This trend reaches its peak during wartime [36].

While Crawford interprets high levels of own-country nominations as indicative of national chauvinism, we take a broader perspective on self-nominations that accounts for their structural role in scientific recognition. As explicitly prohibited by the Nobel Prize Statutes, self-nomination in the strict sense – scholars nominating themselves – does not occur. Instead, we define self-nominations as cases where scientists nominate colleagues from their own organizational or national affiliation, which represents the closest possible form of self-nomination within the given rules.

This practice aligns with broader patterns of academic self-reproduction, such as universities hiring their own PhD graduates, which is particularly prevalent among high-status institutions [37]. It reflects mechanisms of social closure and elite cohesion, where established networks reinforce their own members' visibility and prestige.

Crawford's work [13,36] categorized such nominations as problematic from a universalist perspective. However, this distinction is not always clear-cut: A scientist nominated by fellow nationals may well be a deserving candidate based on scientific excellence, just as a cross-national nomination could be influenced by informal personal networks rather than objective merit. The example of Henri Poincaré illustrates this tension – he was frequently nominated by French colleagues (seemingly a "favourite son"), yet also widely recognized internationally and nominated across multiple countries, including Germany, Italy, Japan, Russia, Sweden, Switzerland, Spain, the Netherlands, the UK, and the USA.

Given these ambiguities, we approach self-nominations as a sociological phenomenon that reflects underlying structures of scientific hegemony. Instead of categorizing them as meritocratic or particularistic, we consider how dominant actors consolidate influence and maintain epistemic control. Patterns of self-nomination can thus be seen as part of broader stratification mechanisms that shape which scientific communities gain international recognition and which remain at the margins. Therefore, we put forward the following hypothesis:

H2: The transition in global scientific leadership from Germany to the United States is reflected in their proportion of self-nominations for the Nobel Prize.

In addition to examining nominations, we also investigate whether nominators are able to secure successful outcomes, which we term "placement power". This concept is derived from studies of graduate placement in academia, as discussed by Clauset et al. [38] and Wapman et al. [37]. It is anticipated that nations with a greater degree of prestige, such as Germany and the United States during their respective periods of scientific hegemony, will demonstrate a higher propensity of advancing successful nominations. Similarly, the success of nominees should be correlated with the scientific prominence of their nation of origin. Therefore, we put forward the following hypothesis:

H3a: The probability of successfully advancing candidates for the Nobel Prize is greater for nominators from nations that wield global scientific leadership.

H3b: The provbability of attaining a Nobel Prize is greater for nominees from nations that wield global scientific leadership.

## 3.  Data & methodology

This study employs a dataset of all Nobel Prize nominations from 1901 to 1969 in the categories of Physics and Chemistry to investigate the impact of shifts in institutional and national structures on global scientific leadership. This approach enables the identification of the ways in which nomination patterns reflect the ascendance and decline of dominant academic centres, thereby emphasizing the enduring interplay between national prestige and scientific authority.

The dataset was derived from the Nobel Foundation's nomination archive [39], which provides nomination data that becomes publicly available after a 50-year embargo. Consequently, the dataset encompasses the period up to 1969 and comprises 8,832 nominations that involve 3,319 individuals. It was curated and enhanced with biographical data of both nominees and nominators, including precise information about the workplaces of nominees and nominators at the time of their nominations. The enhanced dataset permits the tracking of mobility and the provision of a more precise view of

national and organizational affiliations at the time of nominations, representing an advancement over previous studies on Nobel Prize nominations.

While prior research on Nobel Prize nominations has relied on majority rules [8] or rules of thumb [13,36] to assign national affiliations, our approach utilizes the actual country where the scientist was employed at the time of nomination. This ensures a more accurate reflection of mobility, particularly for scientists who migrated as a result of persecution or war, as evidenced by the works of Waldinger [32–34].

We build on previous studies of changes in scientific leadership [29–31], which have tracked the career trajectories of Nobel laureates and already provide well-established insights into long-term institutional prestige and historical patterns, particularly with regard to global competition for scientific talent. However, by examining nominations and not just awards, our study offers a more dynamic perspective on how scientific influence was perceived and distributed at specific points in time - before it was finally consolidated in Nobel prizes.

The entire dataset was subjected to rigorous quality control, adhering to the "four-eyes" principle. All inconsistencies were subjected to a rigorous review, correction, and documentation process. A manual is provided with the dataset, which has been made publicly available to promote transparency and facilitate future research utilizing this resource.

All calculations were performed using Microsoft Excel and R. With regard to our initial set of hypotheses, we describe the evolution of nominations by the national affiliation of both nominators and nominees, with a particular focus on the prominence of Germany and the United States. In order to present numbers of nominees and nominators over time, we use 5-period moving averages, which serve to balance the inherent volatility of nomination patterns with the clarity and precision with which trends over time can be discerned.

To illustrate the centrality of the United States and Germany in the nomination process, we utilize network graphs that display organizational nomination networks. The size of the nodes (representing organizations) is proportional to Kleinberg's hub and authority centrality measures, which capture both the role of central nominating organizations (hubs) and those that receive frequent nominations (authorities). The nodes are coloured according to their national affiliation with concentration on Germany, the United States, and the United Kingdom (other countries are captured in a residual category).

National affiliation of organizations pertains to current borders at the time of the respective nomination. Special consideration is given to organizations with shifting national affiliations: In these cases, organizations appear in our network graphs as two separate entities. To illustrate, the University of Strasbourg is considered German until 1918, after which it is classified as French. The network graphs are generated for two time periods (1901–1933 & 1934–1969) in order to visualize the shift in scientific leadership.

In order to facilitate a comparison between periods of peace and war within the early 20th century, as outlined in hypothesis H2, we employ the use of Phi correlations, which enable us to analyse whether there is a notable increase in the prevalence of self-nominations during one particular period. The Phi value ranges from -1 to 1, with 1 indicating a perfect positive dependence and -1 indicating a perfect negative dependence between two variables. A value of 0 indicates the absence of an association.

In order to test hypotheses H3, inferential statistics are employed to ascertain whether national affiliation of nominees and nominators predicts successful nominations. Logistic regression enables the modelling of the probability of a binary outcome (success or failure) based on one or more explanatory variables. In this study, the binary outcome variable is defined as the awarding of the Nobel Prize by a given nominee in a specific year. A nomination is deemed successful if it results in the Nobel Prize being awarded, with exceptions made for instances where the award is postponed due to war and subsequently conferred the following year.

Period-based models permit a comparison of nominations across different historical phases (e.g., pre- and post-WWII). This method facilitates the capture of temporal dynamics pertaining to shifts in scientific leadership.

Furthermore, our regression models concentrate on a comparison between Germany, the United States, and a residual category comprising all other countries. As a robustness check, the United Kingdom and France are treated as distinct categories to represent nations that have neither experienced an ascent nor a descent in its global scientific leadership position during the observed period. This comparison enables us to contrast the trajectory of a "stable nation" and the changing global scientific dominance of Germany and the United States.

In interpreting the logistic regression coefficients, we focus on their significance and direction, with positive coefficients indicating an increased probability of nomination success. We also calculate average marginal effects (AMEs) to provide more intuitive interpretations. AMEs represent the average change in the probability of success for a one-unit increase in an explanatory variable. To evaluate model fit, we use McFadden's and Nagelkerke's pseudo-R-square measures. Additionally, we employ the Akaike Information Criterion (AIC) and Bayesian Information Criterion (BIC). The BIC is particularly valuable as it penalizes model complexity, helping us achieve a balance between fit and simplicity [40].

The original dataset contained 8,832 nominations, but due to missing values and the need for a 10-year observation period, the sample for regression analysis was reduced to 8,110. The initial ten-year period was characterized by considerable fluctuations, and thus the dataset for regression analysis commences in 1911, when the process had reached a state of stability.

## 4. Empirical results

Germany and the United States are the most prominent countries in terms of both submitting nominators and receiving nominations for the Nobel Prize. Together, these two countries account for nearly 40 percent of all nominators (Germany: 19%, US: 20%; Fig. 1) and 50 percent of all nominations (Germany: 20%, US: 33%; Fig. 2). The next two countries in the Top-4 ranking are France (11% for nominators, 10% for nominees) and the United Kingdom (8% for nominators, 12% for nominees).

The transition in leadership between Germany and the United States during the first half of the 20th century represents a significant finding. At the outset, Germany was the preeminent nation in both the number of nominators and nominees.

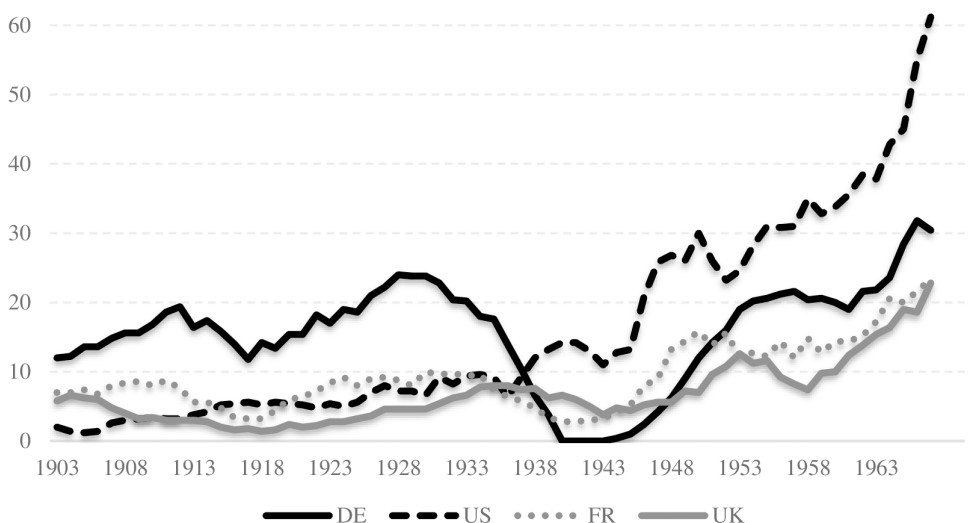

**Fig 1. Top-4 countries with Nobel Prize nominators.** Note: Five-year moving averages are used for the number of nominators. Country abbreviations are given based on the ISO 3166 standard.

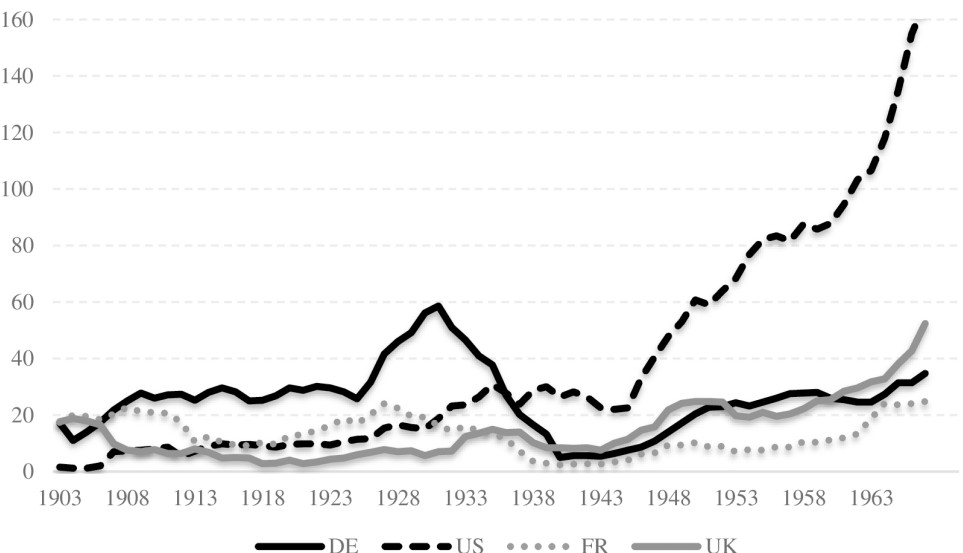

**Fig 2. Top-4 countries receiving Nobel Prize nominations.** Note: Five-year moving averages are used for the number of nominees.

However, its position underwent a precipitous decline during the early 1930s, even before the Nazi regime. During this period, the number of German nominators declined substantially, as a consequence of the Nazi regime's boycott of the Nobel Prize and the forced migration of German scientists. The decline of German scientific dominance was succeeded by the concurrent ascendance of the United States. By the late 1930s, the United States had overtaken Germany, expanding its influence exponentially in the aftermath of WWII. Despite a partial recovery, Germany's position never reached the levels it had held prior to WWII.

The time series data in Figs 1 and 2 provides a visual representation of the global shift in scientific hegemony. During the early 20th century, Germany was the prevailing scientific global power, particularly between 1901 and 1933. Thereafter, the United States assumed a dominant position, which continued until the end of the observation period in 1969. This reflects the broader historical context of the two world wars, the migration of scientists, and the changing political landscapes in both countries.

Fig 3 illustrates the alternation in German and US American scientific leadership, with Germany at the top of both the distribution of nominees (38%) and nominators (31%) during 1901–1933, and the United States leading between 1934–1969. The takeover is evident for nominators with a coverage of 24 percent of the overall distribution since 1934, but is particularly pronounced for nominees, who have received 42 percent of all nominations. The division of the observation period into two relatively equal halves is in close alignment with Germany's political course: The enactment of discriminatory legislation such as the Law for the Restoration of the Professional Civil Service marked a significant turning point across a range of domains, including the general political landscape as well as academia.

For comparison, we also examine the period around WWI to assess whether similar patterns emerge. While political and institutional upheavals affected both periods, nomination data provides a lens on Germany's international scientific standing. It is possible that Germany's nomination counts had already declined significantly due to WWI, meaning that WWII was not the sole driving factor behind its scientific decline.

Dividing the data into two periods – before WWI (1901–1914) and after WWI but before the political regime change in Germany (1919–1933) – shows that the share of German nominators remained stable at around 34% in both periods. This suggests that WWI did not immediately hinder Germany's ability to participate in the nomination process. Among nominees, Germany's share even increased from 31% to 43%, contradicting the notion that WWI already weakened its

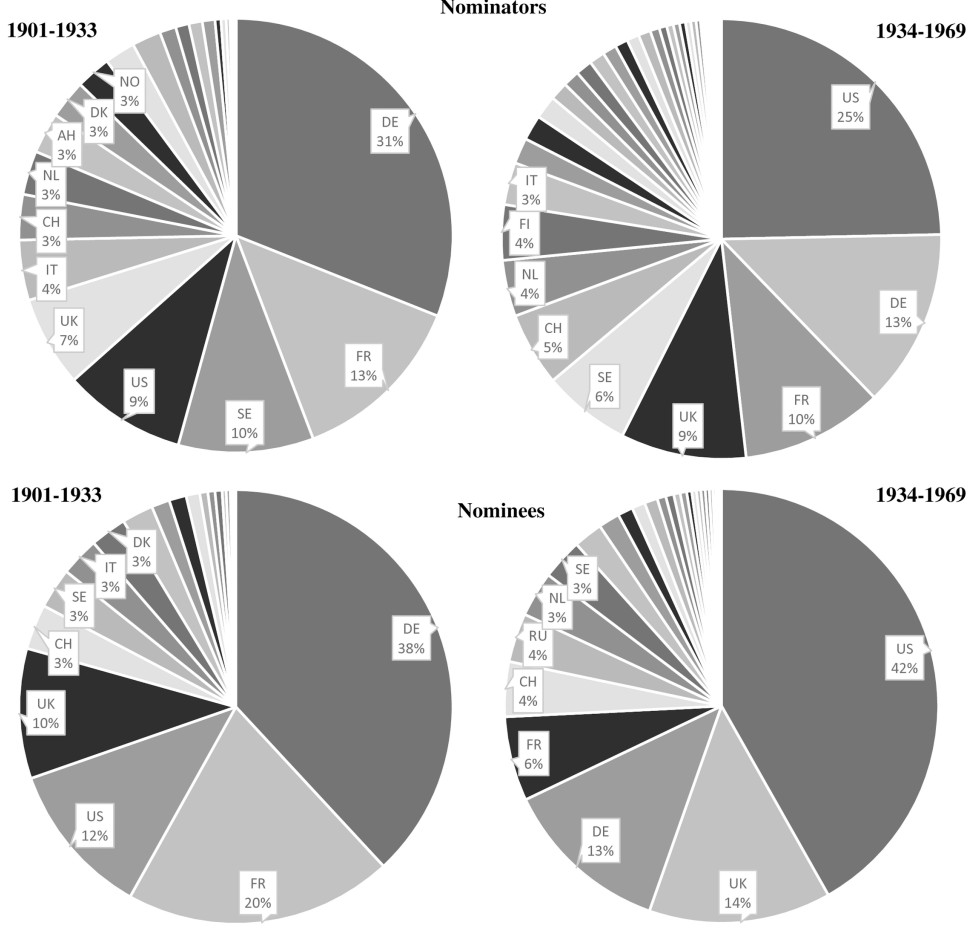

**Fig 3. Proportions of nominators and nominees, 1901-1933 and 1934-1969.** Note: Five-year moving averages are used. Country abbreviations are given based on the ISO 3166 standard.

scientific standing. Meanwhile, in the US, the share of both nominators and nominees nearly doubled, reflecting the country's growing influence in global science.

At the same time, country-specific variations indicate that the effects of WWI were not uniform. In the UK, the share of nominators dropped from 7% to 4%, while the share of nominees declined from 15% to 5%, suggesting a possible war-related setback. France, on the other hand, exhibited more stability, with the share of nominators remaining at 15% and the share of nominees declining only slightly (from 24% to 18%). These patterns suggest that while WWI disrupted scientific exchange in some countries, Germany's role in the nomination process remained resilient.

At the organizational level, evidence suggests that nomination networks have undergone comparable shifts. Prior to 1934, the preeminent scientific centres were predominantly German. After that, universities and other research organizations in the United States began to emerge as significant players on the global scientific stage. In the period following WWII, universities, such as MIT, Caltech and UC Berkeley, became the focal points for nominations, while German universities, such as the HU Berlin, witnessed a notable decline.

These findings are consistent with both hub centrality (organizations frequently nominating) and authority centrality (organizations frequently nominated) measurements captured in Fig 4. In addition to becoming more prominent, US-based organizations benefited from structural advantages, including a diverse funding landscape comprising both public and

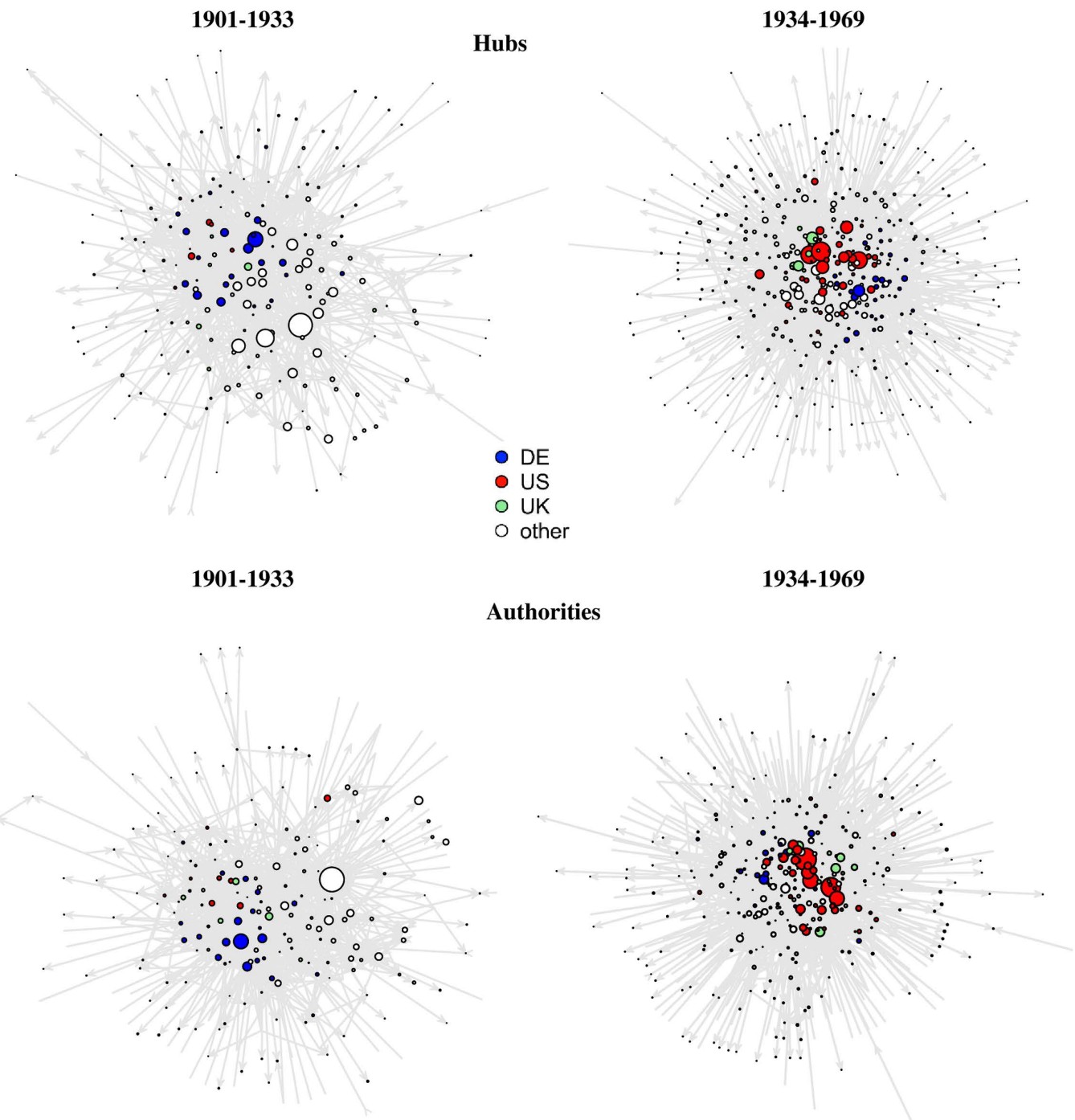

**Fig 4. Organizational nomination network illustrated by hub and authority centrality, 1901-1933 and 1934-1969.** Note: Focus on the main cluster. Nodes are sized according to their hub/authority centrality and coloured according to their national affiliation. The largest white coloured hub as well as authority in the period 1901-1933 is the University of Paris. The size of the University of Paris in relation to the other nodes can be explained by considering that all 13 now autonomous universities in Paris were historically grouped under this label (the division took place in 1970).

private universities, which fostered competition and innovation. In the United States, prominent examples after 1933 include Caltech, UC Berkeley, Princeton University, Rockefeller Institute, Bell Laboratories, Carnegie Mellon University, and Argonne National Laboratory, among others. This suggests that the shift was not solely driven by state-governed universities (in contrast to the situation in Germany), but also by private universities and corporate laboratories. Thus, the structural advantages of the United States are highlighted, as plurality in funding and multiple organizational types foster productive competition within the scientific system [2,24]. At the individual level, numerous US nominators and nominees have a migration history, exemplified by the case of Otto Stern, who fled from Germany to the United States and subsequently held a position at Carnegie Mellon University.

In general, self-nominations show a relatively consistent influence over time, with self-nominations accounting for almost 50 per cent of all nominations. Upon examination of the fluctuations within the historical curves, it becomes evident that peaks in self-nomination proportions do happen in war times, aligning with the findings of Crawford [36]. However, these peaks are not as evident in the overall trend between 1901 and 1969, where fluctuations happen frequently. A comparison of the war years and non-war years reveals no significant difference in the distribution of countries' self-nominations, as indicated by a Phi coefficient of -0.05. This relatively low value serves to diminish the perceived correlation between war and self-nominations at the national level. Specifically, the proportion of self-nominations varies considerably between countries. For example, the United States and France have higher rates than the United Kingdom, in line with Crawford's results [13] .

Fig 5 presents a comparative illustration of self-nomination rates (five-period moving averages) over time for Germany, the United States, the United Kingdom, and France. It is noteworthy that rates were higher during periods of scientific hegemony. Germany's rates reached a peak during its dominant phase, with values ranging up to 90 percent in the 1910s. In contrast, the US saw more stable and high levels of self-nominations following its ascension to global leadership in science.

This indicates that a country's global scientific position may be a contributing factor in the variation of their self-nominations, a finding that contrasts with Crawford's perspective of self-nominations simply as a marker of chauvinism. A comparison of self-nominations in Germany and the United States during the periods of German dominance (1901–1933) and US dominance (1934–1969) provides further support for this interpretation. In comparison to the Phi coefficients for

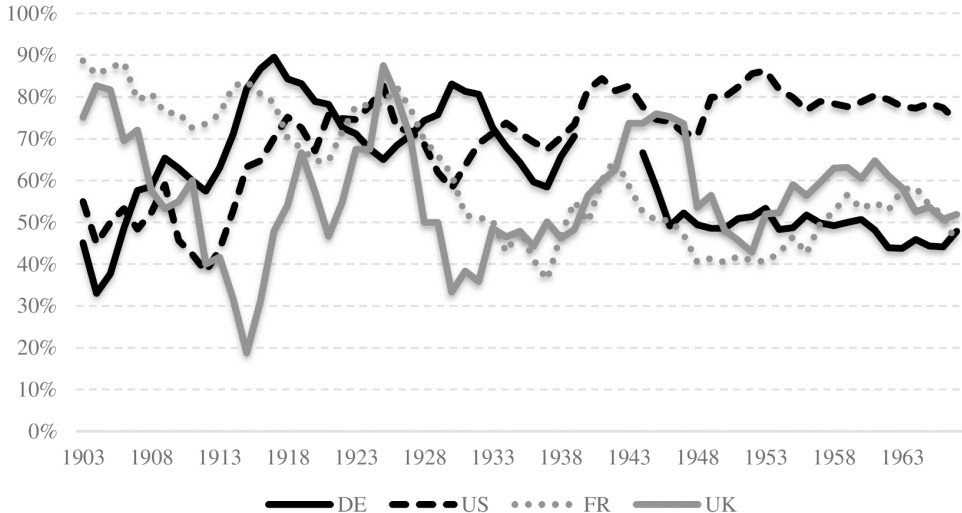

**Fig 5. Self-nominations (as percentage of all nominations).** Note: Five-year moving averages are used.

all other countries within the two specified time periods, which exhibited a mean value of about -0.05, both the US and German coefficients demonstrated a greater degree of volatility: With 0.1 for the USA and -0.2 for Germany, the Phi values are not particularly high. However, they show that self-nominations in Germany decreased considerably in the second period (1939–1969), while they increased in the United States.

Our results indicate a stratification within the scientific community when self-nomination rates at the organizational level (scientists nominated their work colleagues) are considered. It is notable that US organizations, including Rockefeller Institute (51% self-nominations), Caltech (44%) and Princeton (39%), demonstrate high rates of self-nomination. Notably, leading German universities, such as Heidelberg (22%) and HU Berlin (17%), also exhibit elevated self-nomination rates, though they remain much lower than those observed in the US. In contrast, there are examples such as MIT (10%) and LMU Munich (6%) that have very low self-nomination rates, suggesting that within leading countries there is considerable variation in self-nomination behaviour among prestigious universities and institutes.

The following section uses logistic regression models to examine whether national affiliation influences the success of nominations, assuming that the placement power of nominators and the claims of nominees are more successful if they originate from Germany or the United States. The results are analysed across time periods in order to explore potential shifts in the distribution of scientific hegemony, particularly from Germany to the United States.

Table 1 highlights the differences in nominees' success across countries, with Germany and the United States standing out as significantly more successful compared to other countries. The coefficients for these countries show a clear advantage. These positive results confirm that nominees from hegemonic nations are more likely to be successful. However, nominators' placement power shows a different pattern, indicating that nominators from Germany and the United States are less successful in placing their candidates compared to other countries.

**Table 1. Logistic regression model for making a successful nomination.**

| | Coefficient | AME | Lower limit | Upper limit |
|---|---|---|---|---|
| **Nominees'variables** | | | | |
| Country | | | | |
| Reference category: all other countries | | | | |
| Germany | 0.50*** (0.10) | 0.05 (0.01) | 0.03 | 0.08 |
| USA | 0.42*** (0.08) | 0.04 (0.01) | 0.03 | 0.06 |
| **Nominators'variables** | | | | |
| Country | | | | |
| Reference category: all other countries | | | | |
| Germany | −0,39*** (0.06) | −0.04 (0.01) | −0.06 | −0.02 |
| USA | −0.27*** (0.10) | −0.03 (0.01) | −0.05 | −0.01 |
| N | 8110 | | | |
| McFadden Pseudo R² | 0.01 | | | |
| Nagelkerke Pseudo R² | 0.01 | | | |

Logistic regression coefficients and AMEs (average marginal effects) with lower as well as upper limits of the 95% confidence interval. Standard errors in brackets:

*$p < 0.05$,

**$p < 0.01$,

***$p < 0.001$

**Table 2. Logistic regression models (time periods) for making a successful nomination.**

| | 1911-1933 | AME | Lower limit | Upper limit | 1934-1969 | AME | Lower limit | Upper limit |
|---|---|---|---|---|---|---|---|---|
| **Nominees'variables** | | | | | | | | |
| Country | | | | | | | | |
| Reference category: all other countries | | | | | | | | |
| Germany | 0.77*** (0.15) | 0.10 (0.02) | 0.06 | 0.14 | −0.17 (0.17) | −0.01 (0.01) | −0.04 | 0.01 |
| USA | −0.11 (0.26) | −0.01 (0.02) | −0.06 | 0.04 | 0.51*** (0.10) | 0.05 (0.01) | 0.03 | 0.07 |
| **Nominators'variables** | | | | | | | | |
| Country | | | | | | | | |
| Reference category: all other countries | | | | | | | | |
| Germany | −0.43** (0.15) | −0.05 (0.02) | −0.09 | −0.02 | −0.41** (0.15) | −0.04 (0.01) | −0.06 | −0.01 |
| USA | −0.34 (0.28) | −0.04 (0.03) | −0.11 | 0.02 | −0.23* (0.10) | −0.02 (0.01) | −0.04 | 0 |
| N | 1932 | | | | 6178 | | | |
| McFadden Pseudo R² | 0.02 | | | | 0.02 | | | |
| Nagelkerke Pseudo R² | 0.03 | | | | 0.03 | | | |

Logistic regression coefficients and AMEs (average marginal effects) with lower as well as upper limits of the 95% confidence interval. Standard errors in brackets:

*$p<0.05$,

**$p<0.01$,

***$p<0.001$

The AMEs presented point to rather small effects: while German nominees have an average 7 percent higher chance of winning the Nobel Prize than nominees from other countries, nominators from Germany have an average 4 percent lower chance of submitting a successful nomination. As a robustness check, we have added two models to the supplements (S1 Table) to assess whether the effects change when we add the UK and France (otherwise included in all other countries). In principle, these show the persistence of small but significant effects.

In order to capture the historical shift in scientific leadership from Germany to the United States, data is divided into two periods (1911–1933, 1934–1969), analogous to the descriptive analysis (Table 2). As a robustness check, we have added two models with France and the UK to the supplements (S2 Table).

In the first period (1911–1933), nominees from Germany exhibited a markedly elevated success rate in comparison to those from other countries, whereas nominees from the United States did not demonstrate a notable degree of success. In the second period (1934–1969), this situation changes, with US nominees having a significantly higher chance to be successfully nominated. This trend is consistent with the global shift in scientific hegemony. Nevertheless, the negative placement power for German nominators persists across both periods, thus challenging H3a. Similarly, while the placement power of US nominators shows a slight increase, it remains negative overall, indicating that neither country's nominators have a clear advantage in securing successful placements, even as global leadership shifts.

## 5. Summary

Our findings offer insights into the shifting dynamics of scientific leadership and its influence on Nobel Prize nominations over the 20th century. The results corroborate the hypotheses set forth in the theoretical framework of this paper. Moreover, our analysis is consistent with previous research on the redistribution of scientific influence. In particular, our data on nominees and nominators confirm the findings of previous studies on Nobel laureates [28,31]. Urquiola [30], for example,

shows that the US first overtook Germany in the number of mentions in the biographies of Nobel laureates between 1921 and 1940. This period is also central to our study, in which we identify the early 1930s as a crucial turning point in the dynamics of nominations. The convergence of results across different methodological approaches strengthens the robustness of these findings.

H1 proposed that nominations from leading countries (Germany, United States) would have an increased share in periods of global scientific hegemony, a claim that is supported by our data. Results indicate that during the first half of the 20ᵗʰ century, German nominators exhibited significantly higher distribution rates in comparison to their counterparts from other countries. However, beginning in the 1930s, there was a notable decline in the proportion of German nominators, while the share of US nominators increased. This shift is consistent with broader geopolitical changes, as the United States emerged as the dominant global scientific power.

Similarly, scientists from scientifically hegemonic countries are nominated more often during these periods. Results demonstrate a skewed distribution, with US nominees progressively assuming a dominant position in the nomination landscape, particularly during the 1930s. As Germany's role diminished, the United States came to represent a substantial proportion of candidates, thereby further demonstrating a shift in scientific hegemony. The stark contrast between the United States in the early 20ᵗʰ century and in the post-World War II era illustrates the far-reaching consequences of scientific and political shifts on the Nobel nomination process.

H2 is also supported: self-nominations are connected to global leadership in science. Although our data partly supports Crawford's findings, we find no evidence that self-nominations only peaked during wartime on a national level. Instead, we find evidence that fluctuations in nations' share of self-nominations are associated with their scientific hegemonic position. For instance, Germany exhibits a decline in rates following WWII, whereas the United States demonstrates an increase. Furthermore, our analysis indicates that nominators demonstrate considerable support for their compatriots, but scientists from the same institutes and universities also frequently nominate each other. Although fluctuations in self-nominations are volatile, the general pattern supports H2, particularly in reflecting long-term changes.

Finally, our proposition that nominees from scientifically hegemonic countries (Germany, United States) would have an increased likelihood of nomination success (H3b), is partially supported by our data. The findings suggest that during the early decades of the 20ᵗʰ century, German nominees exhibited higher success rates relative to their counterparts from other nations. Similarly, following WWII, the success of German nominees declined, while that of US nominees increased. This shift is consistent with broader geopolitical changes, as the US emerged as the dominant global scientific power. Nevertheless, the influence of national affiliation on success was not as pronounced as anticipated. Therefore, H3b is supported, yet the effect size is small.

Our assumption that nominators from hegemonic scientific countries (Germany, United States) would demonstrate enhanced placement power in securing successful nominations (H3a), is clearly not supported. Although both Germany and the United States retained considerable influence in the nomination process, neither of the two countries exhibited a distinct advantage in securing successful placements over time. Despite the ascendance of the United States, nominators from that country did not exhibit the substantial placement power. Similarly, German nominators witnessed a decline in their capacity to successfully nominate after the 1930s. Therefore, H3a is rejected.

## 6. Discussion

The findings of this study highlight significant shifts in scientific leadership and nomination dynamics, while also indicating avenues for further investigation. As the data set on Nobel Prize nominations is publicly available, it offers an invaluable resource for future research.

Several limitations of this study should be acknowledged. First, our analysis our analysis focuses solely on Nobel Prize nominations in Physics and Chemistry, excluding those in Physiology or Medicine. A broader dataset encompassing all scientific Nobel Prize categories could offer more comprehensive insights and facilitate comparisons across

different fields. Second, the temporal scope of the study is limited by data availability, as the Nobel archives only extend until 1969. As a result, trends and shifts in nomination dynamics beyond this period remain unexplored. Third, while our analysis reveals some notable trends in nomination patterns, it does not include all possible factors influencing the selection process. Factors not captured by our data may include aspects related to organizational prestige or individual career trajectories. These considerations highlight the need for further research employing information-enriched datasets to better understand the scientific reward system.

Our findings provide valuable insights and align with previous literature on scientific hegemony and the impact of WWII. However, as observed, WWI did not significantly affect nomination data, particularly regarding Germany's share before and after the war. This finding should be interpreted within a broader context and supplemented by more robust data. For instance, Iaria et al. (2018) [41] suggest that the consequences of WWI extend beyond the trends captured in nomination data alone. They analysed Nobel nomination data from 1905 to 1945 alongside citation and patent records, concluding that WWI significantly disrupted knowledge flows, especially affecting highly connected scientists. Their results indicate that Germany and Austria-Hungary were disproportionately impacted, leading to a measurable decline in groundbreaking research from these countries. In contrast, our descriptive analysis reveals no immediate decline in German nominations following WWI. This discrepancy may reflect Germany's continued visibility in the nomination process, but the deeper structural impacts on scientific productivity and collaboration could require a more nuanced analysis. Further research is necessary to reconcile these perspectives and to examine how disruptions in scientific networks have influenced long-term shifts in global research output.

This study encourages further inquiry into the intricate processes that underlie one of the most prestigious awards in the field of science. A possible further avenue for investigation would be longitudinal qualitative studies, which could add depth to these findings by examining the personal stories of key scientific figures over time. To illustrate, the individual careers of Emil Warburg and Robert Hofstadter, as case examples from different periods of Nobel Prize history, offer insight into the evolving structure of scientific prestige and nomination dynamics. Warburg, a German scientist and professor at the Friedrich-Wilhelm University of Berlin (subsequently known as HU Berlin), represents the pinnacle of German scientific influence during the inaugural decade of the Nobel Prize awards. His close connections with both the German and Swedish academic elites demonstrate the influence of national and institutional affiliations on nominations during this period. In contrast, Robert Hofstadter, a laureate in Physics in 1961 and a professor at Stanford University, exemplifies the growing dominance of the United States in the post-WWII scientific landscape. Hofstadter, whose own success was inextricably linked with the ascendance of Stanford as a preeminent scientific research organization, exemplifies the manner in which organizational elites and national prestige shaped the trajectory of Nobel nominations in the latter half of the 20th century. His example could be used to demonstrate that laureates frequently nominated other scientists from their own university or institute, thereby reinforcing national and organizational leadership.

Case studies indicate the potential for significant qualitative research in the future, examining the ways in which personal and institutional networks have shaped Nobel nominations across different time periods. Scholars could investigate how factors such as academic mobility, migration, and shifting institutional prestige have influenced both nominations and the broader scientific landscape.

## Supporting information

**S1 Table. Logistic regression model for making a successful nomination.** Standard errors in brackets: * $p < 0.05$, ** $p < 0.01$, *** $p < 0.001$.
(DOCX)

**S2 Table. Logistic regression models (time periods) for making a successful nomination.** Standard errors in brackets: * $p < 0.05$, ** $p < 0.01$, *** $p < 0.001$.
(DOCX)

## Author contributions

**Conceptualization:** Thomas Heinze.

**Data curation:** Marie von der Heyden.

**Formal analysis:** Marie von der Heyden.

**Investigation:** Marie von der Heyden, Thomas Heinze.

**Methodology:** Thomas Heinze.

**Resources:** Thomas Heinze.

**Supervision:** Thomas Heinze.

**Validation:** Marie von der Heyden.

**Visualization:** Marie von der Heyden.

**Writing – original draft:** Marie von der Heyden.

**Writing – review & editing:** Thomas Heinze.

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
