## [Decision Letter · Decision Letter 0]

18 Feb 2025

PONE-D-25-01605Decline of German and Rise of North-American Hegemony in Science. Insights from Nobel Prize Nominations (Physics/Chemistry, 1901-1969).PLOS ONE

Dear Dr. von der Heyden,

Thank you for submitting your manuscript to PLOS ONE. After careful consideration, we feel that it has merit but does not fully meet PLOS ONE’s publication criteria as it currently stands. Therefore, we invite you to submit a revised version of the manuscript that addresses the points raised during the review process. Specifically, the reviewers recommended a few references to be added and to explain better some of the terminology used in the manuscript - especially the research lag in nominations, you must analyze how this might impact your results and how it converses with the findings from Urquiola (2020)'s book. Additionally, I'd like to raise some questions: Germany, unlike the United States, suffered the WW1 structural shock on its infrastructure, something the US didn't suffer, but France and the UK also suffered - if the WW1 shock is persistent, then not only Germany, but also France and the UK should experience decline. But I understand that would be an entirely new hypothesis, so it can be a recommendation for future research; however, besides Crawford's study, is there any further studies that use WW1 as a variable? I noticed that WW1 is referenced only once in the manuscript, and WW2 is referenced more; so, given your dataset and analysis, is it possible to say that WW1 did not play a big role in German decline (which started around 1934, according to your manuscript)? I would recommend against including the Economics Nobel for future research (line 532), because it is technically not a "real" Nobel, and it was established in 1969, when the US-American scientific hegemony was well-established, so it raises another set of questions that might be beyond your research scope (see Avner Offer and Gabriel Sordeberg's *The Nobel factor* ). As additional references, I can suggest recent work by W. MacLeod and M. Urquiola's "Why Does the United States Have the Best Research Universities? Incentives, Resources, and Virtuous Circles" and J. Dittmar and R. Meisenzahl's "The Research University, Invention and Industry: Evidence from German History" - they are optional, but they are also recent work.

We look forward to receiving your revised manuscript.

Kind regards,

Rafael Galvão de Almeida, PhD.

Academic Editor

PLOS ONE

Journal Requirements:

Reviewers' comments:

Reviewer's Responses to Questions

**Comments to the Author**

1. Is the manuscript technically sound, and do the data support the conclusions?

Reviewer #1: Yes

Reviewer #2: Yes

2. Has the statistical analysis been performed appropriately and rigorously? 

Reviewer #1: Yes

Reviewer #2: Yes

3. Have the authors made all data underlying the findings in their manuscript fully available?

Reviewer #1: Yes

Reviewer #2: No

4. Is the manuscript presented in an intelligible fashion and written in standard English?

Reviewer #1: Yes

Reviewer #2: Yes

5. Review Comments to the Author

Reviewer #1: This paper present new data from nominations. It is clear and I think it adds to the literature. At the same time, they paper exaggerates exaggerates the novelty of its findings. This is not necessary because, in fact, the paper has enough to make it interesting. Let me highlight two ways in which this happens.

1) The authors of state that "in contrast to previous methodologies", their " approach utilizes the actual country where the scientist was employed at the time of nomination."

2) More importantly they state that "the transition in leadership between Germany and the United States during the first half of the 20th century represents a significant finding."

In fact, the latter finding was already extensively documented by Urquiola (2020) in a book published by Harvard University Press "Markets, minds, and money". That book documents this transition and does so using data on where scientists were employed and/or trained rather than nationality.

Still, there is new material here. For example the book mentioned above uses awards rather than nominations. There are also new covariates here and careful collection of data.

This is why I am suggesting a minor revision I think that the key thing the paper needs to do is to say how it's findings are different from those in the book. That will make the paper stronger and situate it better in the literature.

Reviewer #2: Thank you for the opportunity to review this manuscript. It is a solid study, but I have some questions and comments

- Drawing on the nomination database, the authors discuss centers and peripheries in science. As I understand them, they use the nominations as a “real-time” perspective, but is this really the case? Do not nominators propose candidates for the award that in most cases did their research decades ago? How can the authors take this time-lag into account?

- State of research: During the last couple of years, Seeman and Restrepo have analyzed the nomination database with a focus on chemistry (e.g. https://pubmed.ncbi.nlm.nih.gov/37204108/) I can also recommend the work by Jacob Habinek on nomination networks in science

(Minor note, perhaps there is more recent work than Hollingsworth, Merton and Ben-David)

- Terminology: I do not see a clear definition of what the authors mean with self-nominations, please clarify. My first assumption would be that it is about scholar X nominating himself, but here the authors, I guess, have another definition. Zuckerman and Crawford talk about “favourite sons” (Poincaré), when a scholar is nominated by scholars who work in the same country, see for example Hansson and Schagen on Sauerbruch in NTM or about Nobel nomination campaigns in Hansson and Schlich in Notes and Records

6. PLOS authors have the option to publish the peer review history of their article (what does this mean? ). If published, this will include your full peer review and any attached files.

**Do you want your identity to be public for this peer review?** For information about this choice, including consent withdrawal, please see our Privacy Policy .

Reviewer #1: No

Reviewer #2: No

---

## [Author Response · Author response to Decision Letter 0]

21 Mar 2025

We have uploaded our response to the reviewers as a Word file as part of this revision.

---

## [Decision Letter · Decision Letter 1]

3 Apr 2025

Decline of German and Rise of North-American Hegemony in Science. Insights from Nobel Prize Nominations (Physics/Chemistry, 1901-1969).

PONE-D-25-01605R1

Dear Dr. von der Heyden,

We’re pleased to inform you that your manuscript has been judged scientifically suitable for publication and will be formally accepted for publication once it meets all outstanding technical requirements.

Kind regards,

Nate Breznau

Academic Editor

PLOS ONE

Additional Editor Comments (optional):

The original editor of the manuscript is not longer available to handle it, so I was asked to step in. I have read the first round reviews and looked at the changes to the manuscript. The reviewers now agree, so I will recommend the paper for publication.

Reviewers' comments:

Reviewer's Responses to Questions

**Comments to the Author**

1. If the authors have adequately addressed your comments raised in a previous round of review and you feel that this manuscript is now acceptable for publication, you may indicate that here to bypass the “Comments to the Author” section, enter your conflict of interest statement in the “Confidential to Editor” section, and submit your "Accept" recommendation.

Reviewer #1: (No Response)

Reviewer #2: All comments have been addressed

2. Is the manuscript technically sound, and do the data support the conclusions?

Reviewer #1: (No Response)

Reviewer #2: Yes

3. Has the statistical analysis been performed appropriately and rigorously? 

Reviewer #1: (No Response)

Reviewer #2: Yes

4. Have the authors made all data underlying the findings in their manuscript fully available?

Reviewer #1: (No Response)

Reviewer #2: Yes

5. Is the manuscript presented in an intelligible fashion and written in standard English?

Reviewer #1: (No Response)

Reviewer #2: Yes

6. Review Comments to the Author

Reviewer #1: (No Response)

Reviewer #2: Thank you for the astute changes. It could be expanded a to further elaborate on the nomination (depend on who was invited nominate)

One detail: It is "Physiology or Medicine" not Medicine or Physiology

7. PLOS authors have the option to publish the peer review history of their article (what does this mean? ). If published, this will include your full peer review and any attached files.

**Do you want your identity to be public for this peer review?** For information about this choice, including consent withdrawal, please see our Privacy Policy .

Reviewer #1: No

Reviewer #2: No

---

## [Editor Report · Acceptance letter]

PONE-D-25-01605R1

PLOS ONE

Dear Dr. von der Heyden,

I'm pleased to inform you that your manuscript has been deemed suitable for publication in PLOS ONE. Congratulations! Your manuscript is now being handed over to our production team.

Kind regards,

on behalf of

Dr. Nate Breznau

Academic Editor

PLOS ONE